# In Vitro Evaluation of Azoxystrobin, Boscalid, Fentin-Hydroxide, Propiconazole, Pyraclostrobin Fungicides against *Alternaria alternata* Pathogen Isolated from *Carya illinoinensis* in South Africa

**DOI:** 10.3390/microorganisms11071691

**Published:** 2023-06-29

**Authors:** Conrad Chibunna Achilonu, Marieka Gryzenhout, Soumya Ghosh, Gert Johannes Marais

**Affiliations:** 1Department of Plant Sciences, Division of Plant Pathology, Faculty of Natural and Agricultural Sciences, University of the Free State, Bloemfontein 9300, Free State, South Africa; 2Department of Genetics, Faculty of Natural and Agricultural Sciences, University of the Free State, Bloemfontein 9300, Free State, South Africa

**Keywords:** *Alternaria alternata*, *Carya illinoinensis*, azoxystrobin, fentin hydroxide, boscalid, pyraclostrobin, propiconazole, disease management, fungicide sensitivity

## Abstract

Black spot disease or Alternaria black spot (ABS) of pecan (*Carya illinoinensis*) in South Africa is caused by *Alternaria alternata*. This fungal pathogen impedes the development of pecan trees and leads to low yield in pecan nut production. The present study investigated the in vitro effect of six fungicides against the mycelial growth of *A. alternata* isolates from ABS symptoms. Fungicides tested include Tilt (propiconazole), Ortiva (azoxystrobin), AgTin (fentin hydroxide), and Bellis (boscalid + pyraclostrobin). All fungicides were applied in 3 concentrations (0.2, 1, and 5 μg mL^−1^). Tilt and Bumper 250 EC containing propiconazole active ingredient (demethylation Inhibitors) were the most effective and inhibited all mycelial growth from up to 6 days post-incubation. The other active ingredients (succinate dehydrogenase inhibitors, organotin compounds, and quinone outside inhibitors) showed 75–85% mycelial growth inhibition. The effective concentration to inhibit mycelial growth by 50% (EC_50_) was estimated for each isolate and fungicide. The overall mean EC_50_ values for each fungicide on the six isolates were 1.90 μg mL^−1^ (Tilt), 1.86 μg mL^−1^ (Ortiva), 1.53 μg mL^−1^ (AgTin), and 1.57 μg mL^−1^ for (Bellis). This initial screening suggested that propiconazole fungicide was the most effective for future field trials test and how these fungicides could be used in controlling ABS disease.

## 1. Introduction 

The production of pecans (*Carya illinoinensis*) in South Africa (SA) currently shows yield limitations mainly due to the development of diseases [1,2]. Alternaria black spot (ABS), caused by *Alternaria alternata*, is one of the more prevalent diseases in SA pecans [3,4]. At present, no ABS disease on pecans caused by species of *Alternaria* has been reported elsewhere in the world. The ABS disease impedes pecan trees in South Africa by causing premature defoliation and poor development of pecan nuts, in turn leading to loss of annual revenue for the industry. This endophytic fungus can occur in pecans without causing any disease symptoms, such as ABS [5], but becomes aggressive and causes disease when the plant encounters temperature, water, or nutrient deficiency stress, similar to those of other reported plants [6,7,8]. *Alternaria alternata* is the most reported species of *Alternaria* [9,10,11,12]. This pathogen has caused various severe global outbreaks on a variety of plants and crops [13,14,15,16], and symptoms on its host develop to form small-circular black spots and eventually progress to a large black lesion on pecan leaves [3,17]. 

Previously, pecan producers in South Africa mainly cultivated crops, such as maize, wheat, and sorghum, and gradually, over the years, transitioned to full production of pecans [18,19]. However, because it was regarded as a relatively small industry till the early 2000’s, pesticide companies were reluctant to invest economically to label fungicides for pecan crops. Therefore, pecan growers have relied on traditional pre-harvest fungicides with a history of efficacy in controlling several other plant diseases, pending labeling on registration for the management of pecan diseases to control pathogens in orchards [20]. Recently, the pecan industry has made efforts to encourage fungicide companies to invest their products in pecans by sourcing potential fungicides approved for other crops, as fungicide application has been a traditional approach to controlling pathogenic fungi, such as *A. alternata* [21]. These fungicides, including azoxystrobin (QoI), fentin hydroxide (OTs), propiconazole (DMIs), pyraclostrobin (QoI), and boscalid (SDHI), are now labeled for pecans [22]. In addition, these fungicides are considered to have a medium to high risk of fungicide resistance due to their single- and multi-site modes of action [23]. Therefore, spraying should only be carried out in mixtures with multi-site fungicides, and repeated spraying with the same mode of action should be avoided [24,25].

Each chemical fungicide or active ingredient has different target sites and modes of action that inhibit the growth of fungal pathogens by targeting specific cell organelles and disrupting their cellular functions [23,26,27], including those selected for this study (Table 1), showing their possible modes of action in the schematic representation (Figure 1). For example, propiconazole is a systemic fungicide and a demethylation inhibitor (DMI) of sterol biosynthesis that disrupts fungal cell membranes by enzymatic oxidation at the side chain attached to the dioxolane ring and by deketalisation with loss of the dioxolane moiety in fungi, thereby blocking demethylation [28]. Azoxystrobin and pyraclostrobin have systemic properties that are part of the group of quinone outside inhibitor (QoI) fungicides that inhibit mitochondrial respiration at the QoI-site of cytochrome *b*, part of the cytochrome *bc1* complex (complex III), which blocks adenosine triphosphate (ATP) production, and eventually inhibit spore germination and mycelial growth in fungal pathogens [29,30]. Boscalid has systemic properties in the group of succinate dehydrogenase inhibitors (SDHI), which target and inhibit succinate-dehydrogenase proteins (complex II) responsible for cellular respiration and energy production in fungi [31]. Unlike systemic fungicides, fentin hydroxide is a non-systemic organotin compound (OTs) fungicide with a multi-site activity that inhibits ATP synthase during the process of oxidative phosphorylation in fungal cell membranes [32]. 

Currently, no extensive research has been conducted to screen for fungicides that could be used against ABS outbreaks in pecans. The aim of this study was to determine the in vitro efficacy of the four fungicides approved for pecans against the pathogen *A. alternata*. Laboratory bioassays were performed using concentrations directly proportional to field application to investigate the potential of these fungicides to inhibit the growth of *A. alternata* isolates. The study expects to establish fundamental information on the potential of these fungicides to be used in the field for future disease management programs. 

## 2. Material and Methods 

### 2.1. Fungal Isolates 

Six *A. alternata* isolates (CGJM3006, CGJM3103, CGJM3078, CGJM3136, CGJM3137, CGJM3142), previously shown to be pathogenic to pecan [3], were used in the in vitro assays. The cultures are maintained in the culture collection of Gert Johannes Marais (CGJM), Department of Plant Sciences, University of the Free State, South Africa. All the fungal isolates were revived on half strength Potato Dextrose Agar (PDA) medium and incubated in a “Labcon LTGC-M40” incubator (Labcon, Gauteng, South Africa) for 14 days at 25 ± 1 °C under 12 h alternating cycles of near-ultraviolet (NUV) light and darkness. 

### 2.2. In Vitro Fungicide Efficacy Assay and the Quantitative Fungicide Sensitivity Estimation Based on EC_50_ Values

Three systemic fungicide products with four active ingredients, namely, azoxystrobin (Ortiva), propiconazole (Tilt), boscalid, and pyraclostrobin (Bellis), and one non-systemic fungicide product, fentin hydroxide (AgTin), were used for the in vitro fungicide test (Table 1). The fungicides selected fall under the group names; quinone outside inhibitors (QoI) (Azoxystrobin), organotin compounds (OTs) (fentin hydroxide), succinate dehydrogenase inhibitors (SDHI) (Boscalid) + quinone outside inhibitors (QoI) (Pyraclostrobin), and demethylation inhibitors (DMI) (Propiconazole), as designated by the Fungicide Resistance Action Committee (FRAC), (www.frac.info), accessed on 20 November 2022). 

The in vitro fungicide assay was performed according to a modified method of Masiello et al. [31]. The concentration of each fungicide was recalculated based on the fungicide dose recommended on the label by manufacturers and diluted with distilled water to four different concentrations, namely, 0 (control), 0.2, 1, and 5 μg mL^−1^ (Table 2). The fungicides were added into autoclaved half-strength PDA medium at 45–50 °C and poured into 90 mm Petri dishes (Lasec, Bloemfontein, South Africa). 

Mycelial plugs (4 mm in diameter), obtained from the margins of actively growing *A. alternata* cultures, were placed at the center of the Petri dishes to corroborate the growth rate. The assay was performed in triplicate. Petri dishes were placed in a “Labcon LTGCM40” incubator (Labcon, Gauteng, South Africa) at 25 ± 1 °C, under an alternating light/darkness cycle of 12 h photoperiod. The inhibitory activity of the fungicides on the colony growth was obtained by measuring the diameter (mm) of developing colonies on the 2nd, 4th, and 6th day of incubation using ImageJ v. 1.53 [33]. The measurements of the mycelial growth rate were performed using a ruler etched in the first image of the control Petri dish as a calibrator to set the scale, and two orthogonal diameter measurements were derived from the ruler in unit per pixels and then converted to mm per length. The diameters for the mycelial inhibitions were measured at both 0° and 90° angles, and the average values of these diameters were further used. Finally, the percentage inhibition of mycelial growth 2, 4, and 6 days post-incubation was determined via the following Equation (1) [34]:(1)$$I=\frac{C-T}{C}\times 100$$
where *I* = Percentage of mycelial growth inhibition; *C* = Mean diameter (mm) mycelial growth of the control colony, and *T* = Mean diameter (mm) mycelial growth of the treatment colonies.

The growth inhibition (*I*) produced by the respective fungicide concentrations was stipulated as a percentage value. This percentage growth inhibition value (%) depicts the difference between maximum mycelial growth with no inhibition on the inoculated Petri dish (control) minus the mean diameter of mycelia growth on PDA Petri dishes with fungicides (treatment) over the mean diameter of control mycelia colony. 

The mycelial growth inhibition data were subjected to a three-way ANOVA analysis of variance using R version 4.1.0 [35] within R-Studio v. 1.3.959 [36] to determine the significant effects of the representative *A. alternata* isolate, fungicide (active ingredient), and fungicide concentration (μg mL^−1^). Means of each isolate for all amended fungicides at day 6 of incubation were compared at a significant level of 0.05 using Fisher’s LSD test function from the “agricolae” [37] and “doebioresearch” (Analysis of Factorial Randomized Block Design for 3 factors) [38] packages, and graphs were plotted with the estimated means using “geom_bar()” function from the “ggplot2” R package v. 3.4.2 [39].

The fungicide concentration that effectively inhibited 50% (EC_50_) mycelial growth for each of six *A. alternata* isolates was estimated by a log-logistic model of three-parameter (LL.3) using the functions “drm” and “estimate_EC_50_()” from the “drc” R package v. 3.0.1 and “ec50estimator” R package v. 0.1.0 [40] to the data over the tested fungicide concentrations for each isolate and percentage mycelial growth inhibition. The model selection procedure was based on the Akaike Information Criterion (AIC) and the “LL.3” with the lower asymptote fixed at zero using the following Equation (2): (2)$$f(x)=\frac{d-0}{1+\exp\left(b\left(\log\left(x\right)-e\right)\right)}$$
where *b* = Rate of decline (slope); *d* = Upper asymptote; 0 = Lower asymptote; *e* = EC_50_, and *x* = Fungicide concentration or dose. 

Mean mycelial growth inhibition percentage and EC_50_ values were subjected to an analysis of variance (ANOVA) (*p* ≤ 0.05), and the statistical significance of differences between the mean results were calculated using Fisher’s LSD test function from the “agricolae” [37] and were visualized with “geom_point” function from the “ggplot2” R package v. 3.4.2 [39].

## 3. Results 

### In Vitro Fungicide Assay 

All the amended fungicide groups showed inhibition against the mycelial growth at the tested concentrations (0.2, 1, and 5 μg mL^−1^), and the inhibitory effects of the mycelial growth were proportional to the fungicide concentrations (Figure 2 and Supplementary Material Appendix A). Contrary to day 6 incubation, there was no clear significant difference (*p* > 0.001) in the percentage inhibition of mycelial growth observed for each fungicide on the second and fourth day (Appendix A). The three-way ANOVA showed significant differences (*p* < 0.001) for the variables “isolate”, “concentration”, and “fungicide”, or the interaction of these variables (Table 2 and Appendix A). All triplicate data at each concentration for the test days were not significantly different (*p* > 0.001), and the mean values were used. Control test at zero concentration differed significantly (*p* < 0.001) with all fungicide treatments, indicating that the fungicide treatments had significant effects. All fungicides with the highest dose (5 μg mL^−1^) were significantly different (*p* < 0.001) relative to the two lower doses at day 6 of incubation. A degree of variation between isolates existed based on Fisher’s LSD test, but all differed significantly from the control. The calculated EC_50_ values for mycelial growth assays for each isolate are listed in Appendix A.

Propiconazole fungicide showed the best inhibitory activity compared to azoxystrobin, fentin hydroxide, boscalid, and pyraclostrobin, as the fungicide completely (100%) inhibited the mycelial growth for all fungal isolates at the two highest doses (1 and 5 μg mL^−1^; *p* = 0. 433) at day 6 (Figure 3A). At the lowest concentration (0.2 μg mL^−1^) for propiconazole, all the isolates were less inhibited, with mean percentage values of 68% (*p* < 0.001), and significantly different from the control. The 50% estimated concentration (EC_50_) values for propiconazole were 1.85 μg mL^−1^ at CGJM3137 isolate to 1.97 μg mL^−1^ at isolate CGJM3006 (Figure 3B), and the overall mean was 1.90 μg mL^−1^ (Table 3). 

Azoxystrobin, as the other active ingredients, inhibited mycelial growth for the concentrations tested (0.2, 1, and 5 μg mL^−1^) on day 6 of incubation but was less effective than propiconazole (Figure 4A). The highest fungicide dose inhibition on the isolates was significantly different (*p* < 0.001) when compared with those of the other two lower doses. The mean percentage values for 5 μg/mL concentration was 82% (*p* < 0.001), 68% (*p* < 0. 001) inhibition values for 1 μg/mL, and at the lowest concentration (0.2 μg/mL); all the *A. alternata* isolates showed inhibition mean percentage value of 55% (*p* < 0.001). The inhibition of the isolates at 0.2 μg mL^−1^ and 1 μg mL^−1^ was significant (*p* < 0.001). All the fungicide doses against each isolate were significantly different (*p* < 0.001) compared to the control. Based on the EC_50_ estimate, the lowest to the highest sensitivity values ranged from 1.48 μg mL^−1^ (CGJM3078) to 2.70 μg mL^−1^ (CGJM3136) (Figure 4B), with an overall mean EC_50_ value of 1.87 μg mL^−1^ (Table 3).

Similar to other active ingredients, fentin hydroxide fungicide showed inhibitory effects on mycelial growth at all concentrations but was less potent compared to propiconazole (Figure 5A). The highest dose (5 μg mL^−1^) inhibited the growth of isolates by an average of 88% and differed significantly (*p* < 0. 001) compared to 0.2 and 1 μg mL^−1^ doses. The lowest concentration (0.2 μg mL^−1^) inhibited the growth of the isolates by an average of 69% and was significantly different (*p* < 0.001). Estimated 50% values ranged from 1.46 μg mL^−1^ for isolate CGJM3142 to 1.61 μg mL^−1^ for isolate CGJM3006 (Figure 5B), with a total average mean value of 1.53 μg mL^−1^ (Table 3).

Boscalid and pyraclostrobin, active ingredients in the commercial product Bellis, inhibited mycelial growth similar to the other active ingredients but were less effective than propiconazole on day 6 of incubation. The highest dose (5 μg mL^−1^) showed a mean percentage inhibition value of 80% and significantly (*p* < 0.001) inhibited the isolates when compared with the two lowest concentrations after day 6 (Figure 6A). 

At 0.2 μg mL^−1^ concentration, the mean percentage values of mycelia growth inhibition were 62% (*p* < 0.001), and the inhibition of the isolates with all the doses differed significantly (*p* < 0. 001). The amended fungicides showed a significant difference (*p* < 0.001) in inhibition compared to the control test. The EC_50_ values for boscalid and pyraclostrobin were 1.51 μg mL^−1^ on isolate CGJM3142 to 1.65 μg mL^−1^ for isolate CGJM3006 (Figure 6B), and a 1.57 μg mL^−1^ overall mean value across the six *A. alternata* isolates (Table 3).

## 4. Discussion 

The current study evaluated the in vitro efficacy of four registered fungicides against six pathogenic *A. alternata* isolates causing ABS disease on pecans. All fungicides showed efficacy on all the isolates at 2-, 4- and 6-days post-incubation period. Propiconazole fungicide, belonging to the demethylation Inhibitors (DMI) group, proved to be the most effective molecule against all the tested isolates, while the other fungicides were less effective. This fungicide significantly inhibited 100% mycelial growth at 1 and 5 μg mL^−1^ and still inhibited at the lowest concentration. 

Previous in vitro fungicide studies showed that propiconazole is effective against *A. alternata* isolated from various crops. Addrah et al. [41] demonstrated a significant reduction in *A. alternata* contamination on sunflower seeds after the application of DMI fungicides. Similarly, the 100% effectiveness of the DMI fungicide assay on *A. alternata*, causing potato brown spot in Afghanistan, has also been reported [42], while Pranaya et al. [43] highlighted that propiconazole strongly inhibited (100%) the mycelial growth of *A. alternata* causing leaf spot of cotton in India. Tilt (propiconazole) fungicide has successfully inhibited the mycelial growth of *A. alternata,* causing leaf spot and fruit rot disease on chili in South Gujarat, India [44]. An additional benefit for pecan producers would be that Tilt, for example, is registered to control other fungi, such as *Cladosporium*, which is the causative agent of pecan scab in South Africa [45]. Fungicide evaluation of Mancozeb, copper-oxychloride, captafol, and propiconazole proved to be effective against *A. alternata* causing leaf blight of groundnut (*Arachis hypogaea*) [46].

Quantitative evaluation of the EC_50_ values showed that propiconazole (DMIs) and azoxystrobin (QoI) were the most efficient fungicides for inhibiting 50% maximal mycelium growth in the investigated isolates compared to other fungicides, corroborating the findings of He et al. [47] and Chitolina et al. [48]. Interestingly, azoxystrobin did not completely inhibit the mycelial growth for isolate CGJM3136 and still showed a high EC_50_ value. This error may not have much practical significance, but it does show the importance of choosing concentrations that will fully inhibit growth and bring the lower asymptote of the dose-response curve to 0% or at least below 50%. However, future studies investigating such isolates could be retested at higher concentrations to test whether a higher EC_50_ value is defined for these isolates and to examine the use of salicylhydroxamic acid (SHAM) to suppress alternative oxidase (AOX) activity, following the procedures described by Ma et al. [49] to test sensitivity to azoxystrobin. 

The target site of all DMIs is the fungal CYP51 enzyme, a cytochrome P450 sterol 14α-demethylase essential for the biosynthesis of fungal sterols [50]. Ergosterol is the most common sterol in fungi because it is an important component of the fungal cell membrane and is essential for fungal growth [51]. However, the resistance to DMIs has been identified in some phytopathogenic fungal species of *Aspergillus*, *Fusarium*, *Rhynchosporium, Penicillium* [52,53,54,55], *Venturia nashicola* [56], *Villosiclava virens* [57], and *Cercospora beticola* [58] for their ability to substitute some amino acids, such as CYP51, in the target protein. Future preliminary studies on the molecular characterization of mutational genes, such as *CYP51A*, *CYP51B*, and *CYP51C*, found in filamentous fungi resulting in amino acid substitutions that alter the structure of the CYP51 protein [59,60,61] are worth investigating as a tool to better elucidate the resistance mechanism of *A. alternata*. 

The other active ingredient groups (SDHI, OTs, and QoI) were not as effective as the DMI group in inhibiting the mycelial growth of all the *A. alternata* isolates tested. All these isolates were able to grow even with the highest dose of the respective fungicides. Previous studies showed that *A. alternata* genotypes possessing H277L or H134R mutations in *sdhB*, *sdhC*, and *sdhD* genes possibly conferred resistance to SDHI fungicides [62]. Furthermore, *A. alternata* mutations with L and R phenotypes carried single- or double-point mutations in *AaSDHB*, *AaSDHC,* and *AaSDHD* genes that encode boscalid target protein [63,64], displaying several amino acid alterations, which, in turn, cause high levels of resistance to boscalid. Similar cross-resistance patterns between two FRAC fungicides (boscalid and pyraclostrobin) were demonstrated in *A. alternata* from pistachio orchards in the USA [63] to quinone outside inhibitors (QoI) [65], and organotin compounds (OTs) fungicides on *A. alternata* from tomato fields in Greece [66] and potato fields in China [47]. 

It is important to note that the experimental laboratory calculations for all the tested fungicide concentrations were proportional to the recommended dose applied in the field by pecan producers [22]. In the field, fungicides are usually applied through spraying programs [67], and the recommended dose of fungicide is 0.005 g/L of water per hectare (10,000 m^2^). This is the equivalent to the laboratory’s highest fungicide dose (5 μg mL^−1^: 5-folds) used in a 90 mm × 15 mm (0.00135 m^2^) Petri dish. Therefore, if *A. alternata* were within the vicinity of a pecan orchard during fungicide application, one could speculate that the dosage would be effective. However, fungicide field tests under various conditions and spraying approaches are needed to test what would be the most optimal and cost-effective. Since environmental factors, such as temperature and precipitation in pecan orchards, may influence the sporulation and growth of *A. alternata* [5,18,19,68], such field trials will test how these in-field conditional variables could interfere with the sensitivity of the fungicides on the pathogen [63,64,69]. Field trials need to be conducted to investigate the application of these selected fungicides to control *A. alternata*. This study demonstrates the in vitro efficacy of the fungicides against the pathogen and creates the opportunity for the pecan industry to further evaluate these fungicides in vivo. For this to happen, the pecan industry would need the collaboration of the fungicide companies to optimally conduct field trials with pecans and take into consideration the basic actions of the fungicides in nature. 

## 5. Conclusions 

This is the first reported in vitro fungicide bioassay on *A. alternata* causing black spot disease of pecans in South Africa. This study provided useful information on the potential efficacy of the six registered fungicides: propiconazole (Tilt); azoxystrobin (Ortiva); fentin hydroxide (AgTin); and boscalid + pyraclostrobin (Bellis). This will enable the pecan industry to consider the fungicides regarding their needs based on effectiveness, economic viability, and practicality of application. The assays suggest that propiconazole fungicides have higher inhibitory effects on *A. alternata*, even at a minimum concentration of 0.2 μg mL^−1^ but are significantly more effective at 1 and 5 μg mL^−1^. For this reason, propiconazole can potentially enhance the control of this fungus. This renders the active ingredient more attractive for the pecan industry due to the potential wider use, ultimately limiting additional fungicide sprays with an economic benefit to the producer. Field trials need to be conducted to determine whether the lowest concentration of propiconazole would still effectively inhibit the pathogen under in vivo conditions, using standard field concentrations, and thus, attempting to avoid unnecessary excessive use of fungicides in the field. 

## Figures and Tables

**Figure 1 microorganisms-11-01691-f001:**
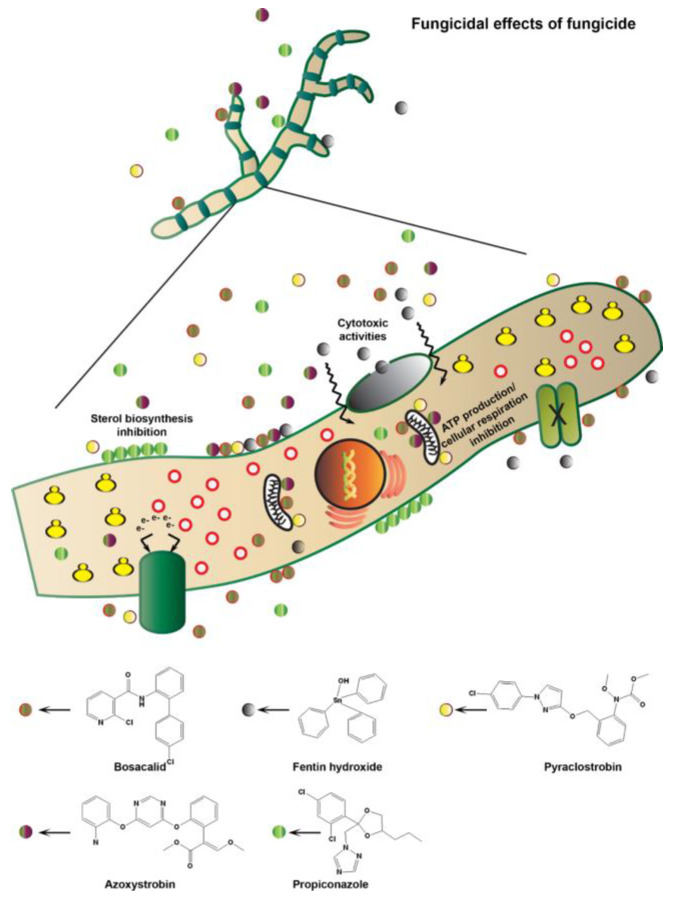
Schematic representation of the fungicide mode of action and how active ingredients inhibit fungal growth.

**Figure 2 microorganisms-11-01691-f002:**
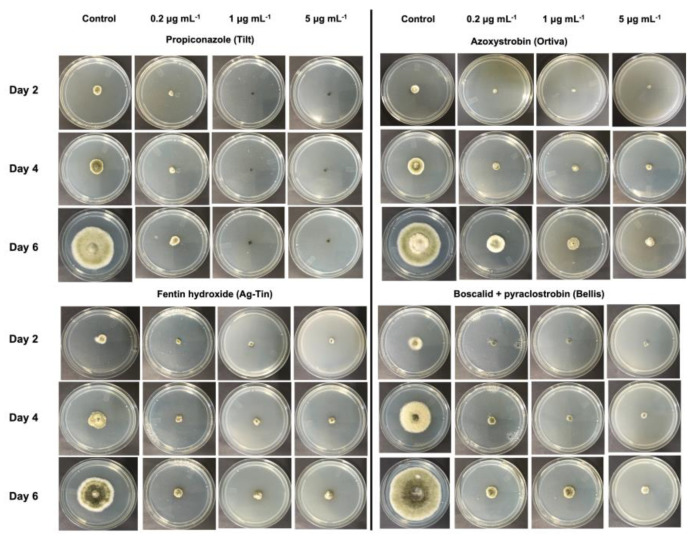
Representation of the effects of different concentrations (0.2, 1, and 5 μg mL^−1^) of propiconazole (Tilt), azoxystrobin (Ortiva), fentin hydroxide (AgTin), and boscalid + pyraclostrobin (Bellis) on *A. alternata* mycelia.

**Figure 3 microorganisms-11-01691-f003:**
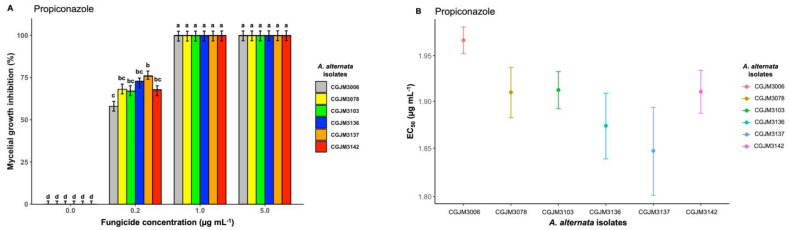
(**A**) Bar plot showing the mycelial growth inhibition percentage of *A. alternata* isolates on potato dextrose agar (PDA) amended with three different concentrations of propiconazole fungicide (expressed in μg mL^−1^); the standard error (SE), represented by error bars, ranged between 0 and 7.5, and different letters on the bars are significantly different. The mean data of the treatments (fungicide concentration) differed significantly from control (Fisher’s LSD test: *p* = 0.05). (**B**) Point plot depicting the effective concentration (EC_50_, μg mL^−1^) values of propiconazole required to inhibit *A. alternata* mycelial growth. The bar heights represent means from three replicates, and standard error bars are shown.

**Figure 4 microorganisms-11-01691-f004:**
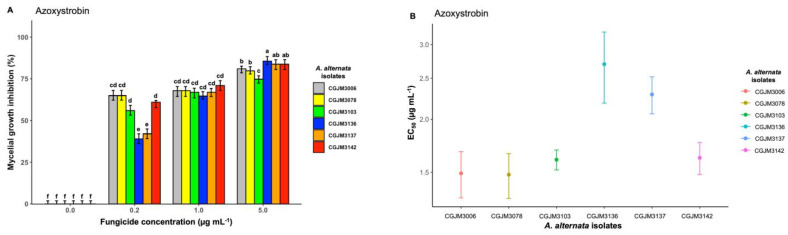
(**A**) Bar plot showing the mycelial growth inhibition percentage of *A. alternata* isolates on potato dextrose agar (PDA) amended with three different concentrations of azoxystrobin fungicide (expressed in μg mL^−1^); the standard error (SE), represented by error bars, ranged between 0 and 7.5, and different letters on the bars are significantly different. The mean data of the treatments (fungicide concentration) differed significantly from control (Fisher’s LSD test: *p* = 0.05). (**B**) Point plot depicting the effective concentration (EC50, μg mL^−1^) values required to inhibit *A. alternata* mycelial growth. The bar heights represent means from three replicates, and standard error bars are shown.

**Figure 5 microorganisms-11-01691-f005:**
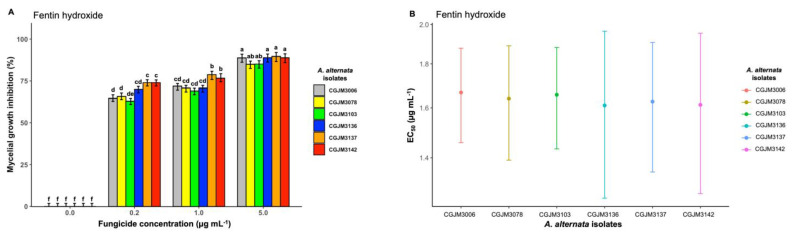
(**A**) Bar plot showing the mycelial growth inhibition percentage of *A. alternata* isolates on potato dextrose agar (PDA) amended with three different concentrations of fentin hydroxide fungicide (expressed in μg mL^−1^); the standard error (SE), represented by error bars, ranged between 0 and 7.5, and different letters on the bars are significantly different. The mean data of the treatments (fungicide concentration) differed significantly from control (Fisher’s LSD test: *p* = 0.05). (**B**) Point plot depicting the effective concentration (EC50, μg mL^−1^) values required to inhibit *A. alternata* mycelial growth. The bar heights represent means from three repeat replicates, and standard error bars are shown.

**Figure 6 microorganisms-11-01691-f006:**
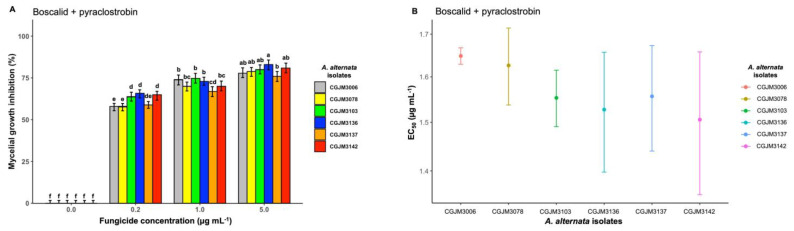
(**A**) Bar plot showing the mycelial growth inhibition percentage of *A. alternata* isolates on potato dextrose agar (PDA) amended with three different concentrations of boscalid and pyraclostrobin fungicide (expressed in μg mL^−1^); the standard error (SE), represented by error bars, ranged between 0 and 7.5, and different letters on the bars are significantly different. The mean data of the treatments (fungicide concentration) differed significantly from control (Fisher’s LSD test: *p* = 0.05). (**B**) Point plot depicting the effective concentration (EC50, μg mL^−1^) values required to inhibit *A. alternata* mycelial growth. The bar heights represent means from three repeat replicates, and standard error bars are shown.

**Table 1 microorganisms-11-01691-t001:** Fungicide product information and their concentration tested on *Alternaria alternata* isolates.

Commercial Name	Active Ingredient	Concen-tration (μg mL^−1^)	Systemic/Non- Systemic	Chemical Group *	FRAC-MoA Group Names *	Target Site *	Mode of Action *	Company Name	Registration Number
Tilt	Propiconazole	0.2, 1, 5	Systemic	Triazole	Demethylation inhibitors (DMI)	C14-demethylase in sterol biosynthesis	Sterol biosynthesis in membranes	Syngenta (Pty) Ltd. Gauteng, South Africa.	L6668
Ortiva	Azoxystrobin	0.2, 1, 5	Systemic	Methoxyacrylate	Quinone outside inhibitors (QoI)	Complex III: Cytochrome bc1 complex	Blocks ATP production	Syngenta S (Pty) Ltd. Gauteng, South Africa.	L5968
AgTin	Fentin hydroxide	0.2, 1, 5	Non- systemic	Triphenytin	Organotin compounds (OTs)	Oxidative phosphorylation, ATP synthase	Cytotoxicity activity	Rolfes Agri (Pty) Ltd. Pretoria, South Africa	L9493
Bellis	Boscalid	0.2, 1, 5	Systemic	Pyridinecarboxamide	Succinate dehydrogenase inhibitors (SDHI)	Complex II: Succinate-dehydrogenase	Respiration	BASF (Pty) Ltd. Gauteng, South Africa	L7817
	Pyraclostrobin			Methoxycarbamate	Quinone outside inhibitors (QoI)	Complex III: Cytochrome bc1 complex	Blocks ATP production		

* Information from Fungicide Resistance Action Committee (FRAC), Mode of Action (MoA). see www.frac.info (accessed on 20 November 2022).

**Table 2 microorganisms-11-01691-t002:** Three-way ANOVA summary showing the interaction effects of *Alternaria alternata* isolate, fungicide (active ingredient), and concentration of fungicide (μg mL^−1^) of day 6 dataset.

Variables	D.f.	Sum Sq.	Mean Sq.	F Value	*p*-Value
Isolate	5	168	34	4525	<0.001 ***
Concentration	3	333,064	111,021	14,941,194	<0.001 ***
Fungicide	3	10,969	3656	492,053	<0.001 ***
Isolate × Concentration	15	284	19	2549	<0.001 ***
Isolate × Fungicide	15	1095	73	9826	<0.001 ***
Concentration × Fungicide	9	7755	862	115,962	<0.001 ***
Isolate × Concentration × Fungicide	45	2370	53	7089	<0.001 ***
Residuals	192	1	0		

Significant codes: “***” 0.001 (*p* < 2 × 10^−16^).

**Table 3 microorganisms-11-01691-t003:** Summary of the overall mean EC_50_ values for each fungicide across all six *Alternaria alternata* isolates.

Fungicide	NoI ^a^	Estimate (μg mL^−1^)	Standard Error	Lower	Upper
Propiconazole (Tilt)	6	1.903490	0.012045937	1.875712	1.931267
Azoxystrobin (Ortiva)	6	1.864728	0.09845003	1.642018	2.087437
Fentin hydroxide (AgTin)	6	1.528238	0.08296495	1.340559	1.715918
Boscalid + pyraclostrobin (Bellis)	6	1.569434	0.04186737	1.474723	1.6641446

NoI ^a^: Number of estimates of EC_50_values, accounting for the number of isolates and repeats. Estimated by a log-logistic model of three-parameter (LL.3) using the functions “drm” and “estimate_EC50()” from the “drc” and “ec50estimator” R packages [40].

## Data Availability

The datasets generated during and/or analyzed during the present study are available from the corresponding author upon reasonable request.

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
