# Peer review of "In Vitro Evaluation of Azoxystrobin, Boscalid, Fentin-Hydroxide, Propiconazole, Pyraclostrobin Fungicides against Alternaria alternata Pathogen Isolated from Carya illinoinensis in South Africa"

_microorganisms, 2023, doi:10.3390/microorganisms11071691_

Round 1

Reviewer 1 Report

Dear authors,

The manuscript entitled "In Vitro Evaluation of Azoxystrobin, Boscalid, Fentin-hydroxide, Propiconazole, Pyraclostrobin Fungicides against Alternaria alternata Pathogen Isolated from Carya illinoinensis in South Africa" presents timely and adequate information to the readers regarding the use of effective fungicides to be used to control ABS disease. The introduction is well wriiten and supported by relevant references. The material and methods are well-described and the results are also clearly presented. The discussion presents also adequate information and is suppported by the results.

Kind Regards

Author Response

Dear authors,

The manuscript entitled "In Vitro Evaluation of Azoxystrobin, Boscalid, Fentin-hydroxide, Propiconazole, Pyraclostrobin Fungicides against Alternaria alternata Pathogen Isolated from Carya illinoinensis in South Africa" presents timely and adequate information to the readers regarding the use of effective fungicides to be used to control ABS disease. The introduction is well written and supported by relevant references. The material and methods are well-described and the results are also clearly presented. The discussion presents also adequate information and is supported by the results.

Kind Regards

Response:

Thank you for your comments and for recommending that this manuscript be accepted for publication.

Reviewer 2 Report

The manuscript presents an in vitro evaluation of a series of fungicides from distinct modes of action against the fungal pathogen Alternaria alternata  from Carya illinoinensis (pecan) in South Africa.

I am recommending accepting the manuscript after major revision, for which I am pointing out minor suggestions of changes in the Introduction and requesting major changes in the Materials & methods, specially considering that the authors did not calculate the EC50 (i.e., the half maximal effective concentration) for each fungicide active and per isolate. EC50 will summarie in a single value the data series of percentage growth in each distinct fungicide doses, which is not meaninful at all to infer differences in efficacy amongst fungicides. Because of this change, results and discussion should be adjusted accordingly. The EC50 is commonly used and recommended by FRAC - Fungicide Resistance Action Committee (check: https://www.frac.info/fungicide-resistance-management/background) to assess fungicide sensitivity shifts in pathogen populations over time. Also check: https://doi.org/10.1094/PDIS-06-17-0873-SR for a thorough on EC50 calculation. 

EC50 can be estimated using the R package ec50estimator coupled with the drc library for analyses of dose response curves, from multiple isolates data sets. You can also test  hypothesis on the most fit model function (fct) for estimating the EC50 by a dose–response curve. Confidence intervals are also estimated. Please check: https://doi.org/10.1614/WT-06-161.1 and https://doi.org/10.3390/agronomy12122952 as references for estimating EC50.

The authors should also indicated if they used salicylhydroxamic acid (SHAM)  to suppress alternative oxidase (AOX) activity, following the procedures described by Ma et al. to test sensitivity to azoxystrobin (check: 10.1128/AEM.72.4.2581-2585.2006).

The minor changes in the Introduction are the following:

Lines 35-36: Describe what types of symptoms A. alternata causes on leveas or other pecan plant tissues. Also, be specific on what types of plant stresses make the symptons more agressive.  

Line 38-39: Fix the phrase accordingly and be specific on where A. alternata causes those black spots: This pathogen [CAUSED?] various severe global outbreaks on a variety of plants and crops, and symptoms on its host devvelop to form small-circular black spots and eventually progress to a large black lesion [ON PECAN LEAVES?]

Lines 44-45: Did you mean companies were reluctant in pursuing the labeling of fungicides for pecan crop, not pecan industry, right? However, because it was regarded as a relatively small industry till the early  2000’s, PESTICIDE companies were reluctant to [invest economically] in TO LABEL FUNGICIDES FOR [pecan industry?] CROPS. 

Lines 47-49: Rephrase for better readability:  Therefore, pecan GROWERS producers have relied UPon traditional pre-harvest fungicides WITH HISTORY OF EFFICACY CONTROLLING SEVERAL OTHER PLANT DISEASE, not L PENDING LABELING registered on FOR THE MANAGMENT OF PECAN DISEASES to manage pathogens in orchards [19]. Recently, the pecan industry has 47 made efforts to encourage fungicide companies to invest their products in pecans by 48 sourcing potential fungicides approved for other crops, as fungicide application is a traditional approach to controlling pathogenic fungi such as A. alternata [20].

Lines 50-53:  Remove all commercial products in brackts as the active ingredients already examplify the fungicides. Rephrase as in the example: These fungicides include, azoxystrobin (Ortiva), boscalid and pyraclostrobin (Bellis), propiconazole (Tilt), fentin hydroxide (Coptin), and propiconazole (Bumper 250 EC), and fentin hydroxide (Ag-Tin) are now [LABELED FOR) pecans [21]. 

Lines 53-54: For accuracy, you need to be specific about which are high or medium risk fungicides for resistance, and how to tacke this problem to avoid the emergence of fungicide resistance in A. alternata, by choosing high-risk only in mixtures with low-risk fungicides. Rephrase as in the example: Azoxystrobin, pyrachlostrobin [quinone outside inhibitors (QoI) fungicides], propiconazole [demethylation inhibitor (DMI) fungicides DMI] fungicides, and boscalid [succinate dehydrogenase inhibitor (SDHI) fungicides]  are considered to have a at medium to high risk FOR FUNGICIDE RESISTANCE due to their single site mode of action [22]. THESE FUNGICIDES SHOULD BE SPRAYED ONLY IN MIXTURES WITH MULTISITE FUNGICIDE, AND REPEATED SPRAYS OF THE SAME MODE OF ACTION SHOULD BE AVOIDED (REFERENCE).

Minor changes recommended.

Author Response

Microorganisms-2395866 In Vitro Evaluation of Azoxystrobin, Boscalid, Fen-tin-hydroxide, Propiconazole, Pyraclostrobin Fungicides against Alternaria alternata Pathogen Isolated from Carya illinoinensis in South Africa”

Reviewer #2

I am recommending accepting the manuscript after major revision, for which I am pointing out minor suggestions of changes in the Introduction and requesting major changes in the Materials & methods, especially considering that the authors did not calculate the EC50 (i.e., the half maximal effective concentration) for each fungicide active and per isolate. EC50 will summaries in a single value the data series of percentage growth in each distinct fungicide doses, which is not meaningful at all to infer differences in efficacy amongst fungicides. Because of this change, results and discussion should be adjusted accordingly. The EC50 is commonly used and recommended by FRAC - Fungicide Resistance Action Committee (check: https://www.frac.info/fungicide-resistance-management/background) to assess fungicide sensitivity shifts in pathogen populations over time. Also check: https://doi.org/10.1094/PDIS-06-17-0873-SR for a thorough on EC50 calculation.

EC50 can be estimated using the R package ec50estimator coupled with the drc library for analyses of dose response curves, from multiple isolates data sets. You can also test hypothesis on the most fit model function (fct) for estimating the EC50 by a dose–response curve. Confidence intervals are also estimated. Please check: https://doi.org/10.1614/WT-06-161.1 and https://doi.org/10.3390/agronomy12122952 as references for estimating EC50.

The authors should also indicate if they used salicylhydroxamic acid (SHAM) to suppress alternative oxidase (AOX) activity, following the procedures described by Ma et al. to test sensitivity to azoxystrobin (check: 10.1128/AEM.72.4.2581-2585.2006).

Response:

Thank you for the comments. We have analysed the EC50 values for each amended fungicide against the tested A. alternata isolate., see updated figures, tables in the manuscripts, and supplementary tables.

The minor changes in the Introduction are the following:

Lines 35-36: Describe what types of symptoms A. alternata causes on leaves or other pecan plant tissues. Also, be specific on what types of plant stresses make the symptoms more aggressive.

Response:

Thank you for the comments. It has been amended, see lines 38 - 39.

Line 38-39: Fix the phrase accordingly and be specific on where A. alternata causes those black spots: This pathogen [CAUSED?] various severe global outbreaks on a variety of plants and crops, and symptoms on its host develop to form small-circular black spots and eventually progress to a large black lesion [ON PECAN LEAVES?]

Response:

Thank you for the comments. All amended, see line 41 and 43.

Lines 44-45: Did you mean companies were reluctant in pursuing the labelling of fungicides for pecan crop, not pecan industry, right? However, because it was regarded as a relatively small industry till the early 2000’s, PESTICIDE companies were reluctant to [invest economically] in TO LABEL FUNGICIDES FOR [pecan industry?] CROPS.

Response:

Thank you for the comments. It has been clarified, see lines 63 - 65.

O Lines 47-49: Rephrase for better readability:  Therefore, pecan GROWERS producers have relied UPon traditional pre-harvest fungicides WITH HISTORY OF EFFICACY CONTROLLING SEVERAL OTHER PLANT DISEASES, not L PENDING LABELING registered on FOR THE MANAGMENT OF PECAN DISEASES to manage pathogens in orchards [19]. Recently, the pecan industry had made efforts to encourage fungicide companies to invest their products in pecans by 48 sourcing potential fungicides approved for other crops, as fungicide application is a traditional approach to controlling pathogenic fungi such as A. alternata [20].

Response:

The comments have been amended, see lines 65 - 67.

Lines 50-53:  Remove all commercial products in brackts as the active ingredients already examplify the fungicides. Rephrase as in the example: These fungicides include, azoxystrobin (Ortiva), boscalid and pyraclostrobin (Bellis), propiconazole (Tilt), fentin hydroxide (Coptin), and propiconazole (Bumper 250 EC), and fentin hydroxide (Ag-Tin) are now [LABELED FOR) pecans [21].

Response:

Thank you for the comments. The above suggestion has been amended, see lines 71 – 72.

Lines 53-54: For accuracy, you need to be specific about which are high or medium risk fungicides for resistance, and how to tackle this problem to avoid the emergence of fungicide resistance in A. alternata, by choosing high-risk only in mixtures with low-risk fungicides.

Rephrase as in the example: Azoxystrobin, pyrachlostrobin [quinone outside inhibitors (QoI) fungicides], propiconazole [demethylation inhibitor (DMI) fungicides DMI] fungicides, and boscalid [succinate dehydrogenase inhibitor (SDHI) fungicides] are considered to have a at medium to high risk FOR FUNGICIDE RESISTANCE due to their single site mode of action [22]. THESE FUNGICIDES SHOULD BE SPRAYED ONLY IN MIXTURES WITH MULTISITE FUNGICIDE, AND REPEATED SPRAYS OF THE SAME MODE OF ACTION SHOULD BE AVOIDED (REFERENCE).

Response:

Thank you for the comments. The above suggestions have been amended, see lines 73 - 76.

Round 2

Reviewer 2 Report

The current version of the manuscript has improved significantly since the authors have considered and incorporated the suggestion from the first-round of reviews. I am now recommending acceptance of the manuscript, after minor reviews are implemented, as suggested:

a) Make sure the authorship of Figure 1 (on the schematic representation of the fungicide mode of actions and how active ingredients inhibit fungal growth) belongs to the authors themselves. If, instead, the Figure was derived from another source, they should refer to the original source and indicate any changes or modifications done.

b) concerning the preferential use of the active ingredient of the fungicides, instead of the commercial names, as this has been a standard in scientific publications alike. Therefore, change figures and tables accordingly as to refer to the specific fungicide active tested: azoxystrobin (instead of Ortiva), fentin hydroxide (instead of AgTin or Coptin), propiconazole (instead of Tilt), boscalid + pyraclostrobin (instead of Bellis), and propiconazole (instead of Bumper 250 EC). Because there was no significant difference between the two commercial fungicides formulated with fentin hydroxide,  the authors could choose to remove one of the treatments or combine the data.

c) Remove Table 2 and transfer the information to a supplementary file, because the data is redundant with data presented in Figures 3, 4 , 5, 6 and 7.

d) Transform contents of the item A from figures 3, 4, 5, 6, 7 in trend lines with each data point at distinct fungicide doses containing the error bars. Fungicide doses should be presented in real scale from 0 to 5, to better represent the trend lines (or even converted in log scale).  Eliminate the statistical comparison of isolates in each dose, which does not make sense, since you have estimated EC50 values, that summarizes the dose response curve for each isolate. You can alternatively choose to compare mean EC50 values of distinct isolates.

Author Response

Reviewer #2-Report-2

The current version of the manuscript has improved significantly since the authors have considered and incorporated the suggestion from the first-round of reviews. I am now recommending acceptance of the manuscript, after minor reviews are implemented, as suggested:

  1. Make sure the authorship of Figure 1 (on the schematic representation of the fungicide mode of actions and how active ingredients inhibit fungal growth) belongs to the authors themselves. If, instead, the Figure was derived from another source, they should refer to the original source and indicate any changes or modifications done.

Response:

Thank you for the comments. Figure 1 is an original image and was not obtained from any online source.

  1. concerning the preferential use of the active ingredient of the fungicides, instead of the commercial names, as this has been a standard in scientific publications alike. Therefore, change figures and tables accordingly as to refer to the specific fungicide active tested: azoxystrobin (instead of Ortiva), fentin hydroxide (instead of AgTin or Coptin), propiconazole (instead of Tilt), boscalid + pyraclostrobin (instead of Bellis), and propiconazole (instead of Bumper 250 EC). Because there was no significant difference between the two commercial fungicides formulated with fentin hydroxide, the authors could choose to remove one of the treatments or combine the data.

 Response:

Thank you for the comments. The names of the active ingredients have been amended instead of their commercial names in the respective figures has been changed and have opted to remove the data of fentin hydroxide (Coptin) and propiconazole (Bumper 250 EC) from the manuscript.

  1. Remove Table 2 and transfer the information to a supplementary file, because the data is redundant with data presented in Figures 3, 4, 5, 6 and 7.

      Response:

Thank you for the comments. Table 2 has been moved to supplementary Table S1

  1. d) Transform contents of the item A from figures 3, 4, 5, 6, 7 in trend lines with each data point at distinct fungicide doses containing the error bars. Fungicide doses should be presented in real scale from 0 to 5, to better represent the trend lines (or even converted in log scale). Eliminate the statistical comparison of isolates in each dose, which does not make sense, since you have estimated EC50 values, that summarizes the dose response curve for each isolate. You can alternatively choose to compare mean EC50 values of distinct isolates.

Response:

Thank you for your comments. However, we prefer to represent Figures “A” (3, 4, 5, and 6) in bar plot and the comparison of each dose and isolate is imperative to the study